# Benchmarking Three Event-Based Rainfall-Runoff Routing Models on Australian Catchments

**David Kemp** * and **Guna Hewa Alankarage**

UniSA STEM, University of South Australia, Mawson Lakes 5095, Australia; guna.hewa@unisa.edu.au
* Correspondence: david.kemp@unisa.edu.au

**Abstract:** In the field of hydrology, event-based models are commonly used for flood-flow prediction in catchments, for use in flood forecasting, flood risk assessment, and infrastructure design. The models are simplistic, as they do not consider longer-term catchment processes such as evaporation and transpiration. This paper examines the relative performance of two widely used models, the American HEC-HMS model, the Australian RORB model, and a newer model, the RRR model. The evaluation is conducted on four case study catchments in Australia. The first two models, HEC-HMS and RORB, do not include baseflow, necessitating the estimation of baseflow through alternate means. By contrast, the RRR model includes baseflow, by extracting a separate loss from the rainfall, and then routing the resultant flow through the catchment, much like quickflow, but with a longer delay time. The models are calibrated and then verified with weighted mean parameter values on an independent set of events in each case study catchment. This gives an indication of the ability of the models to correctly predict flow, which is important when the models are used with design rainfalls to predict design flows. The results demonstrate that all models perform adequately on the four examined catchments, but the RRR model exhibits superior calibration, and, to a lesser extent, better validation compared to the other two models.

**Keywords:** hydrology; models; HEC-HMS; RORB; RRR; flood; runoff; baseflow; benchmarking

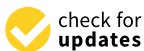



## 1. Introduction

Flooding is a serious and costly type of natural disaster in Australia. For example, the storms and floods that impacted south-east Queensland and coastal New South Wales in February and March 2022 resulted in an estimated insured loss of over AUD 5.56 billion from more than 236,000 claims [1]. However, the economic losses are likely much higher. Flash floods are characterised by their rapid onset (within six hours of rainfall), which leaves very limited opportunity for effective response, making them one of the most hazardous natural events, frequently responsible for loss of life and severe damage to infrastructure and the environment [2]. With a changing climate and increased development of floodplains, managing flood risks is becoming increasingly challenging. Australia is facing a future of more destructive extreme weather events due to climate change, with very wet as well as very dry seasons and weather [1].

Engineers Australia, as the peak body providing guidelines for the assessment of flood risk and infrastructure design, published "Australian Rainfall and Runoff—a guide to flood estimation" in 2019 (ARR2019) [3]. The guide contains two methodologies for estimating flood risk: flow-based techniques using historical streamflow records, and rainfall-based techniques that use historical rainfall information with a transformation to predict flows with the desired probability. Rainfall-based techniques are also known as rainfall-runoff models or runoff routing (RR) models because excess rainfall is routed through a series of conceptual storages to produce an outflow from any rainfall input. As rainfall intensities and stream flow change with climate change, rainfall-based techniques are the only methodology that can be used to predict changes in flood risk. Thus, transformation models must be as robust as possible.

RR models can be physically based or data-driven (for example neural network and statistical) models that use statistical relationships derived from rainfall and river flow data to generate flow forecasts. Generally, these models perform better than others in situations where the underlying interactions and dependencies of physical processes are only partially understood or are unknown [2].

The physical-based models can be event-based or continuous simulation models. Event-based models are commonly used in flood forecasting, especially for short-term predictions, because they focus on individual rainfall events and their impact on the catchment response [4]. These models are also useful in areas where continuous hydrologic data may not be available or reliable. Continuous simulation rainfall-runoff models are designed to simulate the hydrologic response of a catchment over an extended period, typically in the order of years. These models use detailed information on hydrologic processes, such as infiltration, evapotranspiration, and groundwater recharge, and require long-term hydrologic data such as rainfall, temperature, and streamflow [5]. Continuous simulation models are useful for long-term water resources planning, management, and climate change impact assessment.

In Australia, runoff routing models have been used for rainfall-based flood risk estimation for many years. The commonly used RORB model [6] is one example of a runoff routing model. The RORB model, in common with other runoff routing models in Australia, is a single-process model that focuses on surface runoff only. Hence, baseflow must be extracted from the total event flow prior to the modelling and added back to the predicted hydrograph.

Outside Australia, the HEC-HMS model is commonly used to simulate the complete hydrologic process of a catchment [7]. It is a modelling system with a wide choice of methods for the estimation of losses, the transformation function to outflow, and the estimation of baseflow.

Benchmarking is a process where an output or process is compared with a standard, generally accepted to be the best practice or best available. In terms of hydrology, this can be for example the assessment of the performance of the design procedures against at-station historical evidence [8,9], comparison of event-based and continuous simulation [5], assessment of the effect of the number of model parameters [10], or even the comparison of two or more models that have similar structures and outputs [11]. In this study, it is the assessment of the performance of a hydrological model against other models that are identified as best practices.

A review by Kemp and Daniell [12] of flow estimation in Australia concluded that models such as RORB had some limitations, such as the lack of simulation of multiple processes, including baseflow, within the model. The conclusion was made that substantial progress in Australian flood hydrology would be made if a multi-process model was incorporated into general use. The RRR model was developed in the 1990s in Australia as an example of this class of model. However, despite its potential advantages, the RRR model has not been widely applied mainly due to a lack of dedicated software and uncertainty about its performance compared to existing runoff routing models.

To address the uncertainty regarding the performance of the RRR, the model was benchmarked against two models that are industry best practices, namely the HEC-HMS and RORB models. The benchmarking was performed through the calibration of all models on a series of runoff events on four Australian catchments, and the subsequent verification of each model's performance in predicting hydrographs by the application of the calibrated models to an independent set of events using weighted mean parameter values. The performance of each model was then assessed by the application of statistical measures that determine the level of fit to measured hydrographs.

The paper begins with a brief description of each model, followed by the benchmarking assessment procedure and results. We will then provide an analysis of the results and compare the performance of each model. Finally, we will offer some comments on the efficacy of each model based on the results of our study.

## 2. Materials and Methods

### 2.1. Approach to Benchmarking

During the development of the RRR model reported by Kemp [13], the model's performance on rural catchment was assessed by calibrating the model for several events (typically 6) and then applying weighted average parameter values to independent events (typically also 6) for model verification. While this approach provides an absolute measure of the model's performance, it cannot determine if its performance is better than that of any other model. Therefore, to compare the performance of the RRR model against the commonly used RORB and HEC-HMS models, the same procedure was used on Australian case study catchments covering a broader range of locations and climates.

The methodology used in this benchmarking was as follows:

- Four catchments with diverse locations and climates were selected for benchmarking.
- Approximately six events were chosen for each catchment, and each of the three models, RORB, RRR, and HEC-HMS, was calibrated on the events.
- The goodness of fit for all events and models was determined using a non-dimensional statistical measure.
- The RORB model was calibrated only on quickflow, as it does not simulate baseflow. Baseflow was extracted from the total hydrograph.
- Baseflow estimation for HEC-RAS was by the recession of runoff.
- Mean weighted parameter values were calculated for each of the models.
- The measured rainfall and the weighted mean parameter values were then applied to an independent set of approximately six events for each model.
- The estimated baseflow was added back to the RORB-model-estimated hydrograph to obtain the total hydrograph.
- The relative performance of the models was then evaluated using the same non-dimensional statistical measures.

Finally, an overall performance rating was determined for each of the models, based on their relative performance.

The models are described next, followed by a description of the four selected catchments, and a detailed description of the calibration and verification procedure.

### 2.2. The RORB Model

The history of the Australian runoff routing model dates back to the 1960s with the introduction of the Laurenson runoff routing model (LRRM) by Laurenson, in 1964 [14]. The primary objective of the model was to simulate the surface runoff hydrograph of a catchment using rainfall excess while considering the catchment's lag and its variation with discharge.

Laurenson's runoff routing model was a groundbreaking innovation because it introduced the concept of nonlinearity in catchment response. This was a significant departure from earlier models, which assumed that the catchment lag and response time were constant and linearly related to flow. Laurenson's model recognized that catchment response was more complex than previously thought and could vary with catchment flow. As a result, his model was able to capture more accurately the nonlinearities inherent in catchment hydrology, making it a valuable tool for predicting flood flows and designing infrastructure. The LRRM model did not consider the baseflow process in the model.

Following on from Laurenson's initial model, a series of other models with a similar structure were developed, with the calculation of rainfall excess and nonlinear routing of this excess through a series of non-linear conceptual storages. The first version of the RORB program was released as RORT in 1975 [15]. Many further versions have been released since then, but the basic difference between all these and the LRRM model is that the catchment is represented by a distributed network of sub-catchments and channel reaches rather than by isochrones. A detailed description of the RORB model is given in the user manual [6].

Australian Rainfall and Runoff 2019 [3] recommends the use of the Lyne and Hollick filter to extract baseflow from the total observed hydrograph before the RORB modelling. The filtered baseflow is then added to the estimated quickflow hydrograph to obtain the total design hydrograph.

Rainfall is applied at the centroid of each sub-catchment (at A, B, C, D, E, F, and G in Figure 1), and runoff is then calculated by subtracting losses. The losses can be accounted for by using two types of loss models: the initial loss–constant continuing loss (IL-CL) model and the initial loss–proportional loss (IL-PL) model.

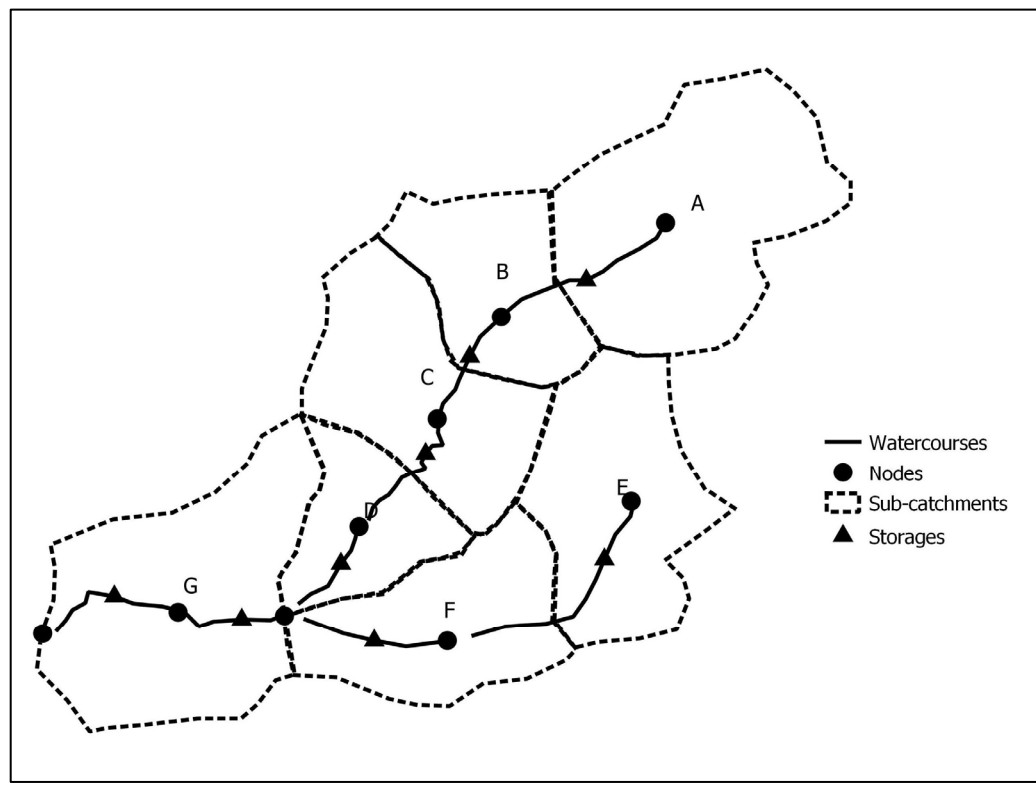

**Figure 1.** RORB node-link model structure.

The resultant hydrograph is then routed through each channel reach storage ($S$) where $S$ is in the form of $S = kQ^m$, and the value of $k$ is determined as $k_c$, an empirical coefficient assigned to the overall catchment, multiplied by the relative delay time ($= \boldsymbol{d}/\boldsymbol{d_{av}}$) where $d$ is the length of the individual storage reach, and $d_{av}$ is the mean flow distance in the catchment.

$$S = 3600 \, kQ^m = 3600 \, k_c \frac{d}{d_{av}} Q^m \tag{1}$$

where

$S$ is the storage (m$^3$);
$k$ is a dimensional empirical coefficient (related to the storage delay time);
$Q$ is the outflow discharge (m$^3$/s);
$m$ is a dimensionless exponent;
$d$ is the length of the individual storage reach (km);
$d_{av}$ is the mean flow distance within the catchment (km).

For this study, we used the initial loss–proportional loss model based on prior research by Kemp and Hewa [16] that demonstrated its superior performance compared to the initial loss–continuing loss model.

### 2.3. The RRR Model

The RRR (rainfall-runoff routing) model was developed in the 1990s [13] but has not been widely used, as no dedicated software was developed until recently. The model represents the catchment as a series of conceptual storages to simulate the flow of water through the catchment. It has ten equal sub-areas with ten equal linear storages representing channel flow. The rainfall excess of each sub-area is added as an input at the sub-area centroids. It is assumed that multiple processes occurring in the catchment can be represented by a separate series of 10 equal sub-areas and 10 non-linear storages representing the hillside runoff processes contributing to each of the 10 channel inflow points. Each runoff process is assumed to follow a different path. For each process, losses are extracted from the total rainfall to provide rainfall excess. An initial loss (IL) is used followed by a proportional loss (PL).

Kemp [13] found that there were generally up to three runoff processes in evidence, and it can be assumed that these represented the following:

1. Baseflow. This is the traditional concept of baseflow and is what is generally referred to as the steady-state regional groundwater runoff; it is the slowest flow process contributing to the hydrograph. It is known that the lag between rainfall and groundwater runoff to the stream discharge can be substantial, due to the long flow path length in the groundwater system.

2. Slowflow, being interflow or throughflow, which can also be labelled as capillary fringe flow. This mechanism acts with a lag from rainfall to stream flow that is less than that of the baseflow, due to the quicker response time from rainfall to runoff into the stream.

3. Fastflow, most probably similar to Hortonian overland flow, either from a part of the catchment area or the full catchment area. The response time of this mechanism is short compared with the two above, as no infiltration and flow through the soil and rock flow is involved.

Each storage process path has a series of ten equal areas and ten equal storages with an excess rainfall input of the form:

$$S = 3600 \, k_p Q^m \tag{2}$$

where

$S$ is the storage (m$^3$);
$k_p$ is a dimensional empirical coefficient (related to the storage delay time);
$Q$ is the outflow discharge (m$^3$/s);
$m$ is a dimensionless exponent.

There will be a separate value of $k_p$ for each process, and these will be labelled by a process number as above. For example, for three processes they are labelled $k_{p1}$, $k_{p2}$ and $k_{p3}$, and losses are also labelled by process number. Each process can have an initial loss (IL1, IL2, and IL3) and a proportional loss (PL1, PL2, and PL3).

At each channel inflow point, there is a storage in the channel of the form:

$$S = 3600 \, kQ \tag{3}$$

where

$S$ is the storage (m$^3$);
$k$ is a dimensional empirical coefficient (related to the storage delay time);
$Q$ is the outflow discharge (m$^3$/s).
The channel storage is thus linear (storage delay time not varying with the flow).

### 2.4. The HEC-HMS Model

HEC-HMS is a hydrological modelling platform that was developed by the Hydraulic Engineering Center (HEC) of the US Army Corps of Engineers (USACE), which is funded

to develop and maintain the software to meet the various needs of practitioners in the USA. HEC-HMS is one of the world's best-studied rainfall-runoff models with an abundance of facilities and a range of linear and non-linear transformation options for subareas and reaches. It includes advanced analysis features and semi-automated parameter optimisation [7].

HEC-HMS contains alternative loss, transformation, and baseflow options. As this study focuses on event models, the following modelling options were used:

- Account of losses using the initial loss–continuing loss model (IL-CL), as the initial loss–proportional loss (IL-PL) model is not available in HEC-HMS.
- Direct runoff transformation by using the Clark Unit hydrograph approach, as this has been shown to give similar results to the RORB model [7].
- Baseflow estimation by the recession of runoff as this is similar to the approach that we adopted to extract baseflow before RORB modelling and is not a rainfall-based baseflow analysis.

By using these options, the HEC-HMS model most closely matches the approach used by RORB, and thus a direct performance comparison can be made.

### 2.5. Case-Study Catchments

Four catchments were selected for this study to represent different climatic conditions across Australia, and thus test the models on a wide range of catchments. To ensure that catchment size did not affect the results, the selected catchments were ideally less than 40 km$^2$ in size. In addition, the pluviometer and flow records were examined to ensure that the chosen catchments had reliable data. The selected catchments included Inverbrackie Creek in South Australia, Finch Hatton Creek in Queensland, Marrinup Brook in Western Australia, and Burra Creek in southern New South Wales. For each catchment, where possible, approximately 12 storm events (6 for calibration and 6 for validation) that produced the highest flows within the period of record were identified and extracted. An exception was made for the Marrinup Brook catchment in Western Australia, where only winter storms were selected. This is because south-west Western Australia catchments have a significant change in response between winter and summer storms [17] and choosing only winter storms ensured consistency in the modelling.

The catchment locations are shown in Figure 2, and the attributes are summarised in Table 1. Though all these catchments had good pluviometer and flow records and represented different climate zones, the catchment areas of two of the selected catchments are slightly over the upper limit of 40 km$^2$.

**Table 1.** Catchments selected for comparison.

| Catchment Name | Station Number | State | Catchment Area (km$^2$) | Station Latitude | Station Longitude | Mean Annual Rainfall (mm) |
|---|---|---|---|---|---|---|
| Burra Creek | 410774 | NSW | 68.7 | −35.54 | 149.23 | 600 |
| Inverbrackie Creek at Craigbank | A5030508 | SA | 8.44 | −34.95 | 138.93 | 810 |
| Finch Hatton Creek | GS125006A | Qld | 35.7 | −21.11 | 148.65 | 2180 |
| Marrinup Brook | 614003 | WA | 45.6 | −32.70 | 115.97 | 1230 |

A summary of the features of each catchment is given next, including a climate classification in accordance with Stern [18]. Land use descriptions are in accordance with the Australian Agricultural and Resource Economics and Sciences classification [19].

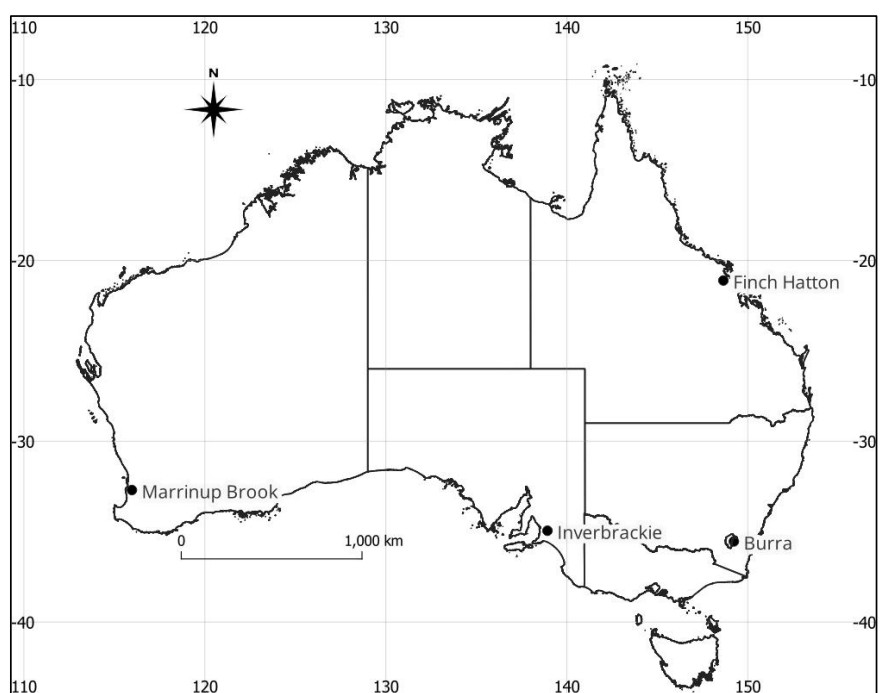

**Figure 2.** Locality plan for selected Australian catchments.

Figure 3 shows the mean monthly rainfall and mean daily maximum temperature for each catchment, together with the standard deviation of both.

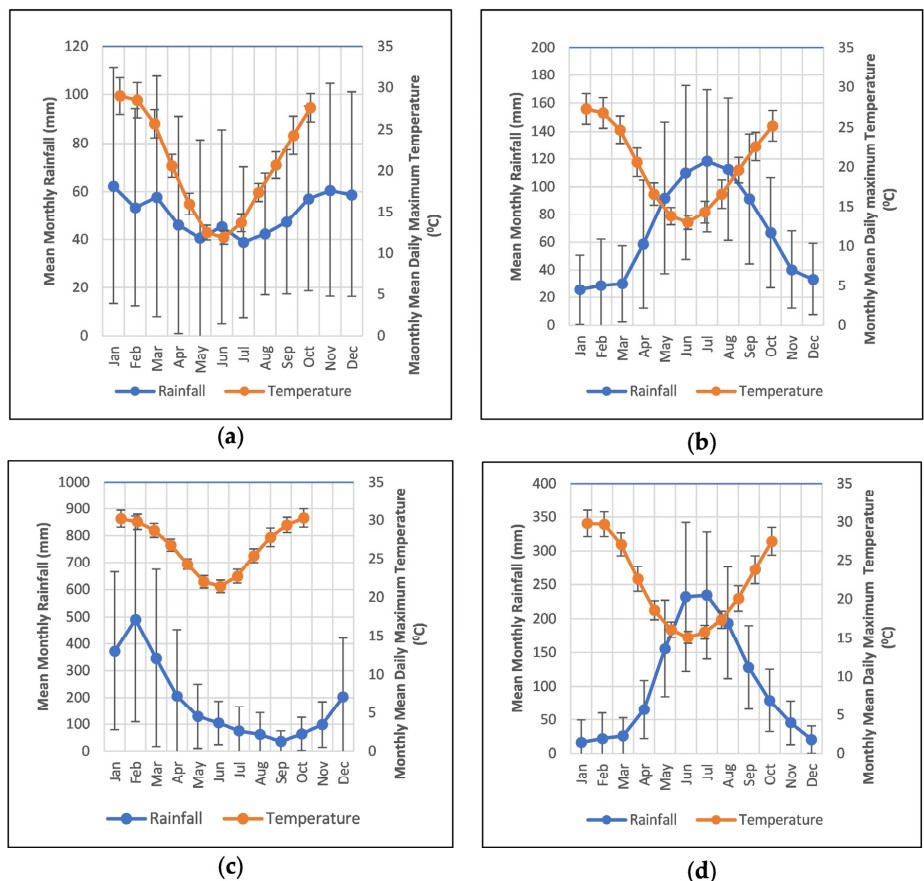

**Figure 3.** Mean Monthly Rainfall and Maximum Daily Temperature with Standard Deviation for Burra Creek (**a**), Inverbrackie Creek (**b**), Finch Hatton Creek (**c**), and Marrinup Brook (**d**).

- Burra Creek catchment

Figure 4 shows the Burra Creek catchment. Burra Creek is a tributary of the Murrumbidgee River, located in south-eastern New South Wales, just south of Canberra. The catchment covers 68.7 square kilometres, encompassing a landscape of gently rolling hills, a mix of open grazing land, and native forest. The creek flows through the catchment, serving as a crucial source of water for agricultural irrigation and supporting the local ecosystems. The catchment is characterised by around 40% hilly and forested (nature conservation) terrain, with the remaining land primarily utilised for grazing. Elevations range from 850 m to 1100 m. The region is classified as having a temperate climate with no dry season.

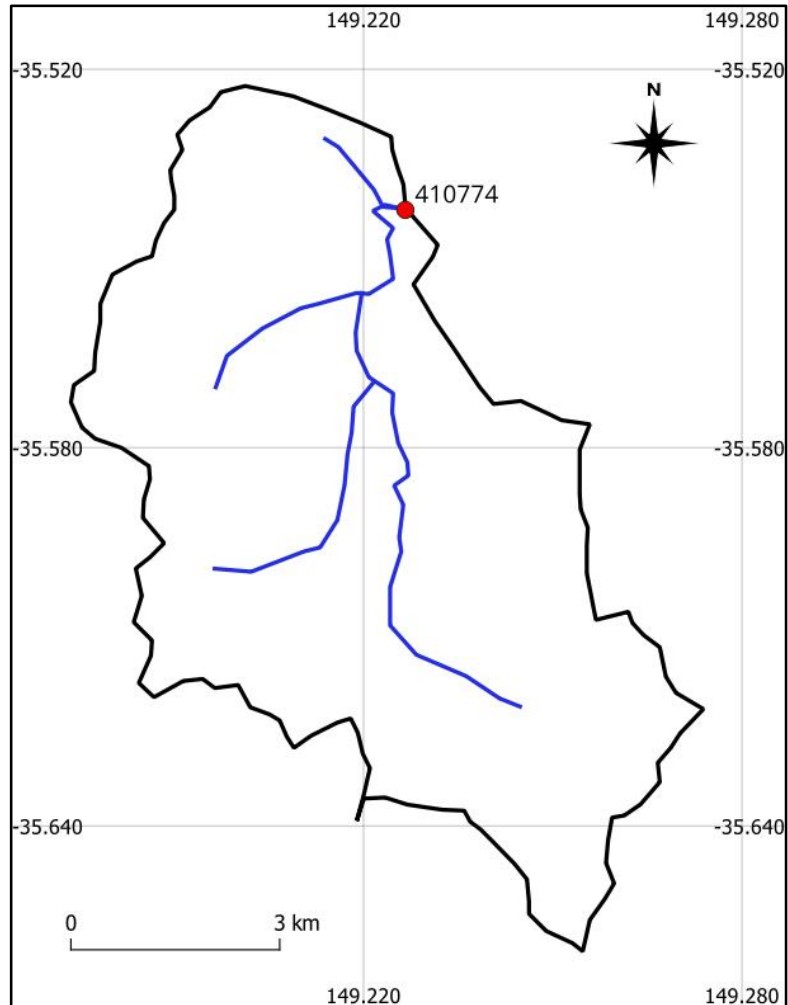

**Figure 4.** Burra Creek catchment.

For the investigation, 12 flood events were extracted from the streamflow records from 1998 to 1995. The extracted events have an average recurrence interval (ARI) varying from 1.5 years to 12.5 years. Seven events were used for calibration, and five for verification. The highest ARI of the calibration events was 9 years, while the highest ARI of the verification events was 12.5 years.

The catchment has a mean annual rainfall of 604 mm, runoff of 63.9 mm, and potential evapotranspiration of 1090 mm.

- Inverbrackie Creek catchment

Figure 5 shows the Inverbrackie Creek catchment. Inverbrackie Creek rises to the north and east of Woodside in the southern Mount Lofty Ranges of South Australia and flows west into the Onkaparinga River south of Woodside. The catchment covers 8.44 square

kilometres and has a single station recording both rainfall and stream flow. The region is moderately hilly, with an elevation range from 430 m to 530 m. The main land use in this small catchment is livestock grazing (44%), with dairying (24%) and horticulture, mainly vineyards (18%), being the other significant land uses, along with vineyards, urban living areas, forests, and areas of remnant native vegetation.

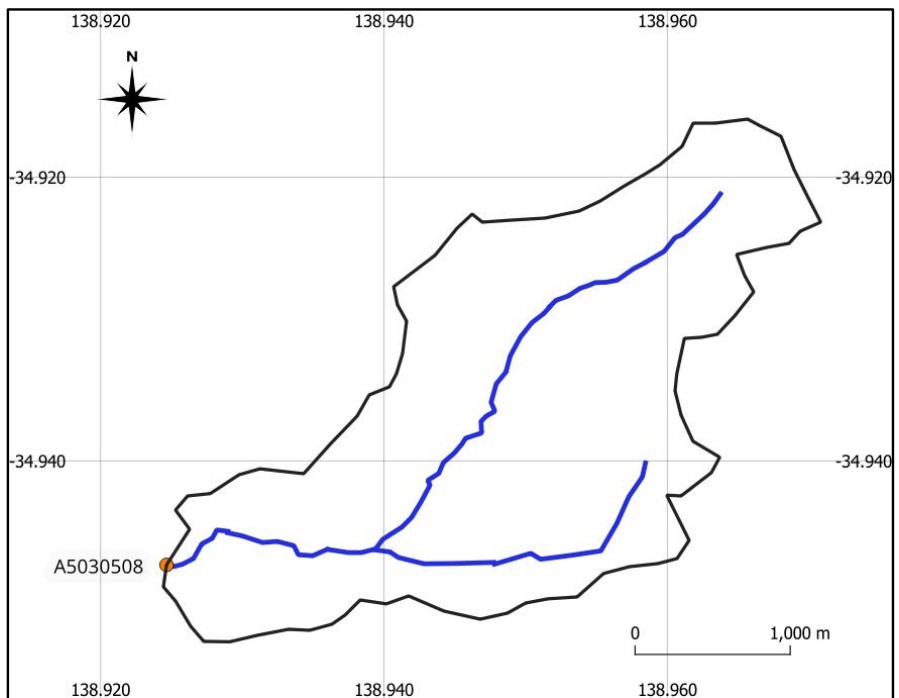

**Figure 5.** Inverbrackie Creek catchment.

For this investigation, a total of 12 flood events were extracted from the streamflow records spanning from 1987 to 1996. The extracted events have an average recurrence interval (ARI) ranging from 0.4 years to 11.3 years. Six events were used for calibration, and the remaining ones were used for verification. The highest ARI of the calibration events was 11.3 years, while the highest ARI of the verification events was 4.7 years. The region has a temperate climate with a distinct dry summer season.

The catchment has a mean annual precipitation of 807 mm, runoff of 9.3 mm, and potential evapotranspiration of 1106 mm.

- Finch Hatton catchment

Figure 6 shows the Finch Hatton catchment. The Finch Hatton catchment lies within tropical north Queensland and covers an area of 35.7 km², with a single station measuring stream flow and rainfall. With steep terrain, the catchment has an elevation range from 100 m to 1150 m within the catchment length of 10 km. The annual rainfall varies significantly within the catchment, with the top of the catchment receiving an estimated 2180 mm compared to 1660 mm at the gauging station. The catchment has some grazing, but the majority is nature conservation, managed resource protection, or other minimal use. The region has a subtropical climate with no distinct dry season.

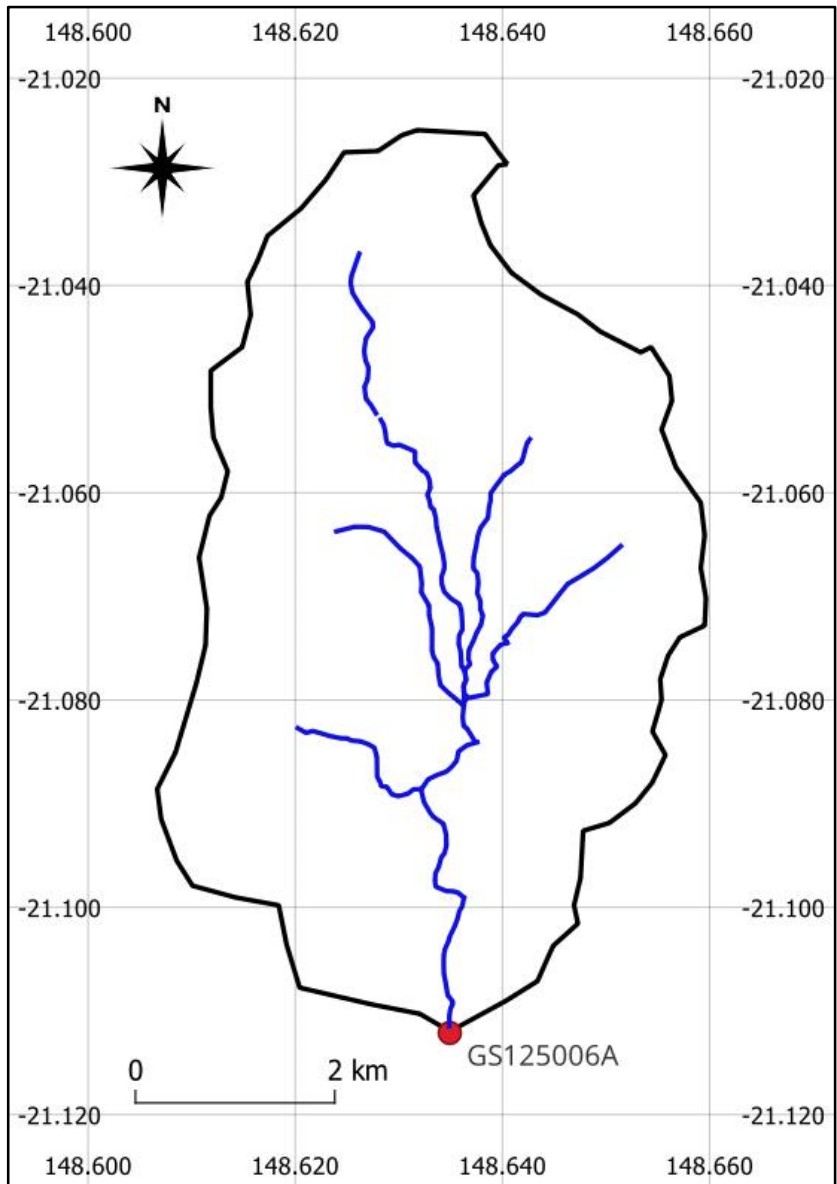

**Figure 6.** Finch Hatton catchment.

For the investigation, a total of 12 flood events were extracted from the streamflow records spanning from 1990 to 2017. Six flood events were used for calibration, ranging from a 5-year ARI to a 150-year ARI. Five events were used for verification, ranging from a 3.8-year ARI to a 7-year ARI.

The catchment has a mean annual runoff of 1508 mm and potential evapotranspiration of 1755 mm.

- Marrinup Brook catchment

Figure 7 shows the Marrinup Brook catchment. The Marrinup Brook catchment is in the south-western region of Western Australia, covering an area of 45.6 km². A single station measures both flow and rainfall within the catchment. The catchment is undulating, with elevations ranging from 80 m to 320 m, and is primarily covered with forests.

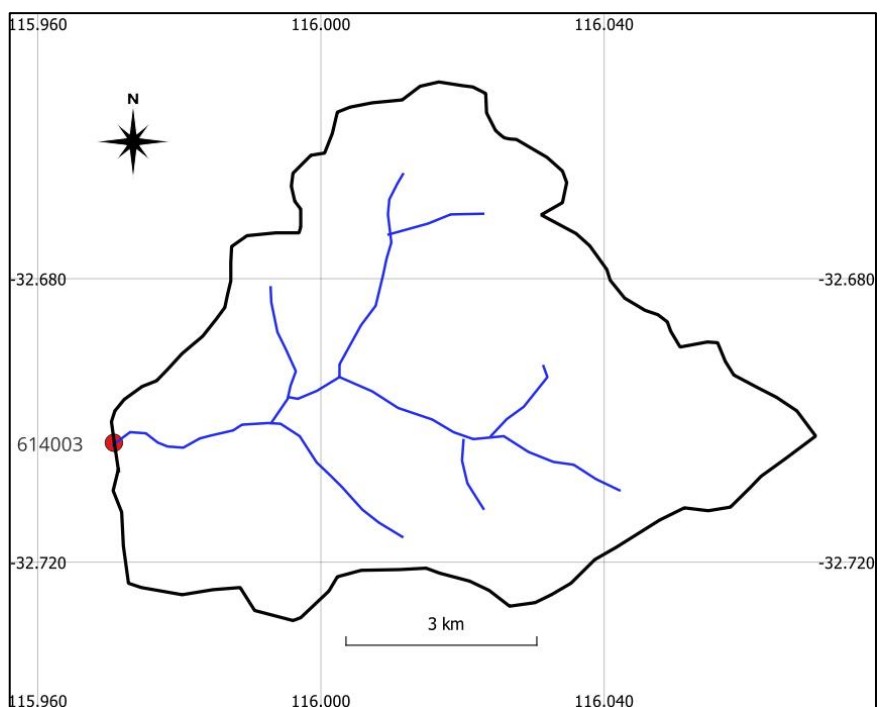

**Figure 7.** Marrinup Brook catchment.

The catchment has a small amount of dryland cropping but is mainly production native forest and other minimal use. The catchment is known for its diverse flora and fauna, including various wetland and riparian habitats and several threatened or endangered species, such as the Western Swamp Tortoise and Carnaby's Black-Cockatoo. For the investigation, six flood events between 1978 and 1992 were used for calibration, ranging from a 9-year ARI to a 23-year ARI. Additionally, six events were used from the same period for verification, ranging from a 1.8-year ARI to an 11.7-year ARI. The region has a temperate climate with a distinct dry summer season.

The catchment has a mean annual rainfall of 1225 mm, runoff of 24.4 mm, and potential evapotranspiration of 1321 mm.

*2.6. Statistical Analysis*

We have chosen the Nash–Sutcliffe efficiency (NSE) [20] and the absolute peak flow error (APFE) to assess the performance of the models. The NSE is defined by Equation (4).

$$NSE = 1 - \frac{\sum_{t=1}^{T}\left(Q_m^t - Q_o^t\right)^2}{\sum_{t=1}^{T}\left(Q_o^t - \overline{Q}_0\right)^2} \tag{4}$$

where

$\overline{Q}_0$ is the mean of observed discharges during the period;

$Q_m^t$ is the modelled discharge at time t;

$Q_o^t$ is the observed discharge at time t.

The percentage absolute peak flow error (APFE) is defined as the absolute value of the ratio of the estimated to measured peak flow, given as a percentage.

These two measures were chosen because they are dimensionless, allowing for comparison of the level of fit across different events. This is not achievable using statistical indicators such as RMSE (root mean square error), MBE (mean bias error), and MAE (mean absolute error).

### 2.7. Baseflow Extraction and Model Calibration

As mentioned earlier, RORB is a single-process model that focuses solely on surface runoff (quickflow). Consequently, effectively addressing baseflow becomes a significant concern during the calibration and verification. In this study, the approach recommended in Australian Rainfall and Runoff 2019 [3] was adopted, involving the separation of baseflow from the total hydrograph before the RORB modelling. The separation was achieved using a 9-pass Lyne and Hollick filter [21], as described in Hill et al. [22].

For most catchments and events, a 15 min time step was utilized along with a Lyne and Hollick filter parameter value of 0.981. This value corresponds to a filter parameter of 0.925 when considering an hourly time step, aligning with the ARR 2019 recommendation. However, due to Marrinup Brook's considerably longer response time, a 1 h time step was employed when modelling this catchment.

The RRR and RORB models were calibrated to achieve the best fit to the observed hydrographs. In the RRR model calibration, the least-squares difference between the predicted and observed hydrographs was minimized, while in the RORB model calibration, the average absolute ordinate error was minimized as it was given as an output for model runs. Both RRR and RORB aimed to maximize the Nash–Sutcliffe efficiency (NSE). By contrast, the HEC-HMS model integrates the objective function to maximize the NSE, eliminating the need for users to define it as required in RORB and RRR.

### 2.8. Model Verification

After calibrating the models for a series of events, a mean value for storage and loss parameters was required to be applied with the models for verification events.

To enable greater weight for better calibration hydrograph fit, the individual event parameter values were weighted by a measure for the goodness of fit for each individual event (a weighting factor) before the mean parameter values were determined. The NSE could not be used for this, as the range was only from 0 to 1.0. Instead, a weighting factor (WF) was calculated for each event by dividing the event observed peak flow by the calibration root mean square error, as defined by Equation (5). The error in the hydrograph fit is thus normalized. A better fit will give a higher weighting factor. Note however that a perfect fit (zero error) will yield an undefined WF. This did not happen in this study.

This approach enabled the calculation of weighted mean values for both the storage and loss parameters. Subsequently, model verification was carried out using the weighted mean parameter values on a set of independent events.

$$WF = \frac{Observed\ peak\ flow}{Root\ mean\ square\ error} = \frac{Q_{op}}{\sqrt{\frac{\sum_1^n (q_0 - q_c)^2}{n}}} \tag{5}$$

where

$WF$ Is the weighting factor for the calibration event;
$q_0$ is the observed flow at each time step;
$q_c$ is the calculated flow at the time step;
$n$ is the number of time steps or observations;
$Q_{op}$ is the observed peak flow.

### 2.9. RORB Model—Baseflow Treatment for Verification

The RORB model can predict only one runoff process from a rainfall input, termed quickflow. To this must be added baseflow to determine the total hydrograph. Baseflow for the verification events was determined using the procedures outlined in ARR2019 [3]. A regional method was developed in ARR2019 to characterise the contribution of baseflow, based on three parameters:

- Baseflow peak factor;
- Baseflow Volume factor;
- Baseflow under peak factor.

These factors are available from the ARR2019 data hub. Using these factors and the time to peak of the hydrograph a complete baseflow hydrograph can be determined. For the purpose of this investigation, the hydrograph used for the estimation of event baseflow was the predicted RORB hydrograph. This estimated baseflow was added to the RORB hydrograph to give the total verification hydrograph.

## 3. Results

The performances of the three models during both the calibration and verification processes are compared separately.

### 3.1. Comparative Performance of the Model Calibration

Tables 2 and 3 provide a comparison of the performances of the three models for all events used in the model calibration, based on NSE and percentage absolute peak flow error (APFE). The best values appear in shaded cells. In addition, the tables display the overall mean of these two statistics for each study catchment, as well as the total number of events that achieved the best result. If two models have the same NSE, this is counted for both models, which may result in a total number of events with the best results that is higher than the total number of events used.

**Table 2.** Calibration results—comparison of model performance using mean NSE.

| Catchment Name | Number of Events Used | No. Passes RORB Baseflow | Mean NSE RORB | Mean NSE RRR | Mean NSE HEC-HMS |
|---|---|---|---|---|---|
| Burra Creek | 7 | 9 | 0.955 | 0.979 | 0.968 |
| Inverbrackie Creek at Craigbank | 6 | 9 | 0.913 | 0.962 | 0.940 |
| Finch Hatton Creek | 5 | 5 | 0.886 | 0.931 | 0.926 |
| Marrinup Brook | 6 | 9 | 0.922 | 0.949 | 0.906 |
| Overall mean for all catchments | | | 0.919 | 0.955 | 0.935 |
| Total number of events with the best result | | | 1 | 17 | 7 |

**Table 3.** Calibration results—comparison of model performance using mean APFE.

| Catchment Name | Number of Events Used | No. Passes RORB Baseflow | Mean APFE RORB | Mean APFE RRR | Mean APFE HEC-HMS |
|---|---|---|---|---|---|
| Burra Creek | 7 | 9 | 14.3% | 3.7% | 6.8% |
| Inverbrackie Creek at Craigbank | 6 | 9 | 14.0% | 6.9% | 6.7% |
| Finch Hatton Creek | 5 | 5 | 12.6% | 7.6% | 9.5% |
| Marrinup Brook | 6 | 9 | 14.3% | 25.8% | 28.8% |
| Overall mean for all catchments | | | 13.8% | 11.0% | 13.0% |
| Total number of events with the best result | | | 7 | 10 | 7 |

During the calibration process in the Marrinup Brook catchment, it was discovered that the RRR model yielded the best results when assuming that only one process was occurring. The results presented in this study reflect this assumption.

To evaluate the overall performance of the hydrological models, the study adopted a statistical framework developed by Ritter and Muñoz-Carpena [23]. This framework classifies the goodness of fit into four performance classes, as shown in Table 4. Similar classification methods can be found in the literature, such as those used by Masseroni and Cislaghi [24], Singh et al. [25], and Hossain et al. [5].

**Table 4.** Rating classes for goodness-of-fit tests.

| | |
|---|---|
| Very good | NSE > 0.75 |
| Good | 0.65 < NSE < 0.75 |
| Satisfactory | 0.50 < NSE < 0.65 |
| Unsatisfactory | NSE < 0.50 |

The level of fit for calibration of all events on every catchment was rated as very good in this study. In summary, all three models showed excellent performance in calibration, with the RRR model consistently producing the best results. The results of the APFE were more evenly spread across the models, as the models were not specifically calibrated for peak flow but for the overall hydrograph, as measured by the NSE.

*3.2. Verification*

Tables 5 and 6 show the results of the verification, using the same two measures of model performance that were used in the calibration, NSE and APFE. As previously stated, these two measures are non-dimensional, and can thus be used to directly compare the performance of the models across all the events modelled.

**Table 5.** Verification results—comparison of model performance using mean NSE.

| Catchment Name | Number of Events Used | Mean NSE RORB | Mean NSE RRR | Mean NSE HEC-HMS |
|---|---|---|---|---|
| Burra Creek | 5 | 0.029 | 0.179 | 0.118 |
| Inverbrackie Creek at Craigbank | 6 | 0.553 | 0.725 | 0.465 |
| Finch Hatton Creek | 5 | 0.729 | 0.796 | 0.710 |
| Marrinup Brook | 6 | 0.704 | 0.755 | 0.096 |
| Overall mean for all catchments | | 0.504 | 0.614 | 0.347 |
| Total number of events with the best result | | 4 | 16 | 2 |

**Table 6.** Verification results—comparison of model performance using mean APFE.

| Catchment Name | Number of Events Used | Mean APFE RORB | Mean APFE RRR | Mean APFE HEC-HMS |
|---|---|---|---|---|
| Burra Creek | 5 | 26.1% | 7.4% | 2.3% |
| Inverbrackie Creek at Craigbank | 6 | 9.1% | 35.5% | 4.2% |
| Finch Hatton Creek | 5 | 15.2% | 24.7% | 19.7% |
| Marrinup Brook | 6 | 14.3% | 25.8% | 28.8% |
| Overall mean for all catchments | | 16.2% | 23.3% | 13.7% |
| Total number of events with the best result | | 10 | 9 | 3 |

As with the calibration, the Nash–Sutcliffe efficiency (NSE) was chosen as the most appropriate criterion to evaluate and compare the overall performance of the three models, based on the statistical framework developed by Ritter and Muñoz-Carpena [23]. The results are shown in Figure 8 and indicate that the RRR model outperforms the other two models with the highest number of very good fits, followed by the RORB model. Conversely, the HEC-HMS model has the highest number of unsatisfactory fits.

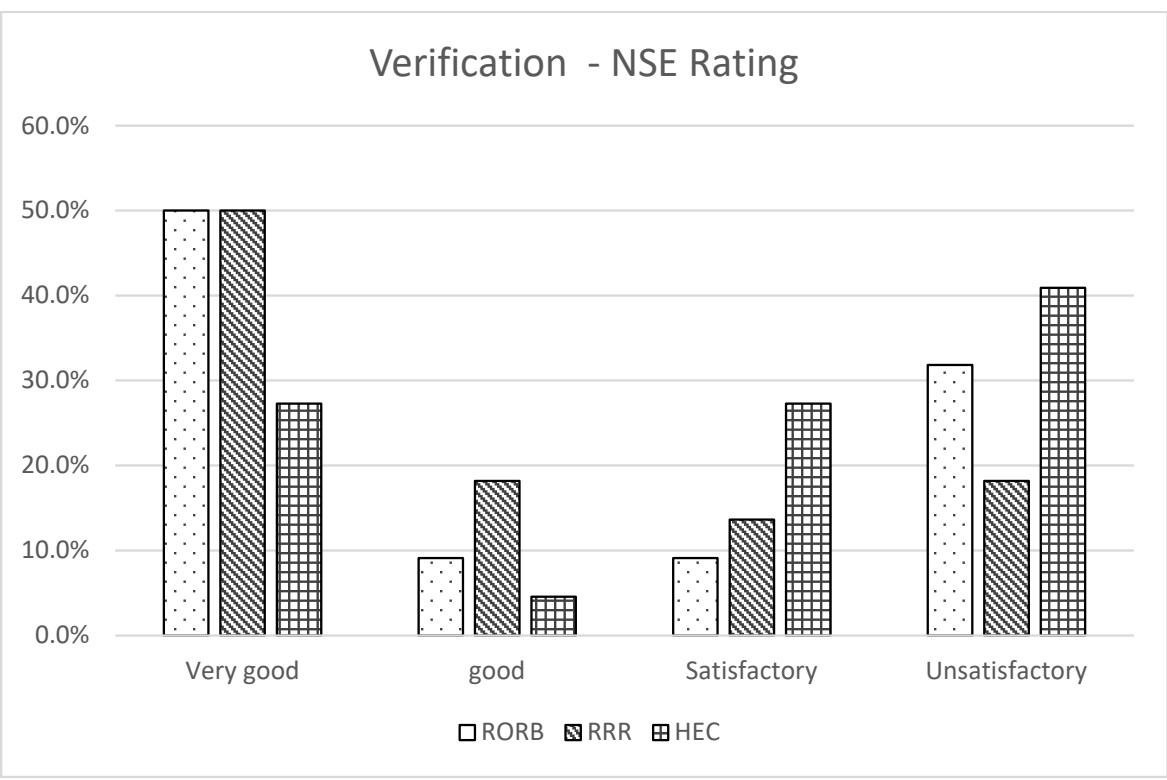

**Figure 8.** Percentages of events that fall into NSE rating classes developed by Ritter and Muñoz-Carpena [23] for the verification of the RORB, RRR, and HEC-RAS models.

## 4. Discussion

The main objective of this study was to compare the performance of three event-based hydrological models for predicting the flow hydrographs of independent events. The significance lies in ensuring the robustness of rainfall-based techniques, as they are the only methods capable of predicting changes in flood magnitude due to climate change.

### 4.1. Model Performance

The calibration results indicate that all three models provided a satisfactory fit to the events and thus performed satisfactorily. However, the RRR model outperformed the others in terms of mean NSE and mean APFE, and also demonstrated superior results for a higher number of events. During the verification phase, the performances of the three models varied considerably, with the RRR model still maintaining the best mean NSE and the highest number of events with the best NSE. On the other hand, the HEC-HMS model had the lowest mean APFE.

It is worth noting that the level of fit, as measured by both NSE and APFE, was generally better during calibration than during verification for all models. This can be attributed to two reasons. First, the calibrated losses compensate for the difference between the actual average catchment rainfall hyetograph and the single rainfall hyetograph used for calibration. Second, during verification, the use of weighted mean parameter values cannot fully account for catchment antecedent conditions. This effect is more pronounced in a drier climate or regions with greater seasonal variation in rainfall or temperature, where catchment antecedent conditions have a greater impact on verification NSE.

Previous investigations comparing these models are scarce. RORB usage is primarily limited to Australia, and the RRR model has not been previously benchmarked against either RORB or HEC-HMS. A study by Jacobs and Ryan [7] only compared the different hydrograph generation options in HEC-HMS.

The RRR model differs from the RORB and HEC-HMS models in that it estimates the baseflow using an empirical runoff routing model based on catchment rainfall, while

the other models use recession analysis of the hydrograph to estimate baseflow without considering catchment rainfall. Despite all three models producing reasonable calibration fits, the verification fits of the RRR model are superior in terms of NSE and rating.

Kemp and Daniell [8] noted that the inclusion of baseflow estimation within the runoff routing model is crucial for advancing the use of such models in Australia. The findings of this study support this argument, as the RRR model with its incorporated baseflow estimation outperforms the RORB and the HEC-HMS models with a simple baseflow assessment.

### 4.2. Baseflow Extraction

In Australia, the extraction of baseflow from a measured rural runoff hydrograph is typically carried out using a recursive digital filter for single-process runoff routing models such as RORB. However, Ladson et al. [26] point out the limitations of this approach, stating that the derived series do not reflect underlying physical processes in terms of shape, timing, or magnitude, making quantitative inferences challenging. They also highlight the substantial variation in estimated baseflow indices when using different software packages, reducing confidence in the approach. This means that comparisons of baseflow indices between studies and over time are difficult and this reduces confidence in the approach. We suggest the Lyne and Hollick filter is still useful but a standard approach to its application is required. A standard application of the filter as suggested by Ladson et al. is not recommended in the ARR2019 guidelines.

### 4.3. Runoff Processes and Monte Carlo Simulation

The RRR model allows for the modelling of multiple processes. Studies by Rogger et al. [27,28] and Basso et al. [29] demonstrate the unexpected occurrence of very large events when catchment storage capacity is exceeded, leading to changes in runoff processes. Single-process event models such as RORB and HEC-RAS cannot adequately model the full range of events expected in catchments. Therefore, the use of Monte Carlo simulation based on parameters determined for normal events as recommended in ARR2019 is not suitable for catchments exhibiting such changes in behaviour.

### 4.4. Effect of Climate on Calibration and Verification Success

Upon analysing the results, a possible relationship between the catchment's average annual rainfall and verification NSE was observed among the catchments examined in this study. Figure 9 shows NSE plotted against the catchment average annual rainfall for each model. However, due to the limited number of catchments analysed in this study, it is not possible to conclusively establish a relationship. It is worth noting that the HEC-HMS model exhibited a notably poor verification fit for one event in the Marrinup Brook catchment, resulting in variations between the three models in the relationship between NSE and mean annual rainfall.

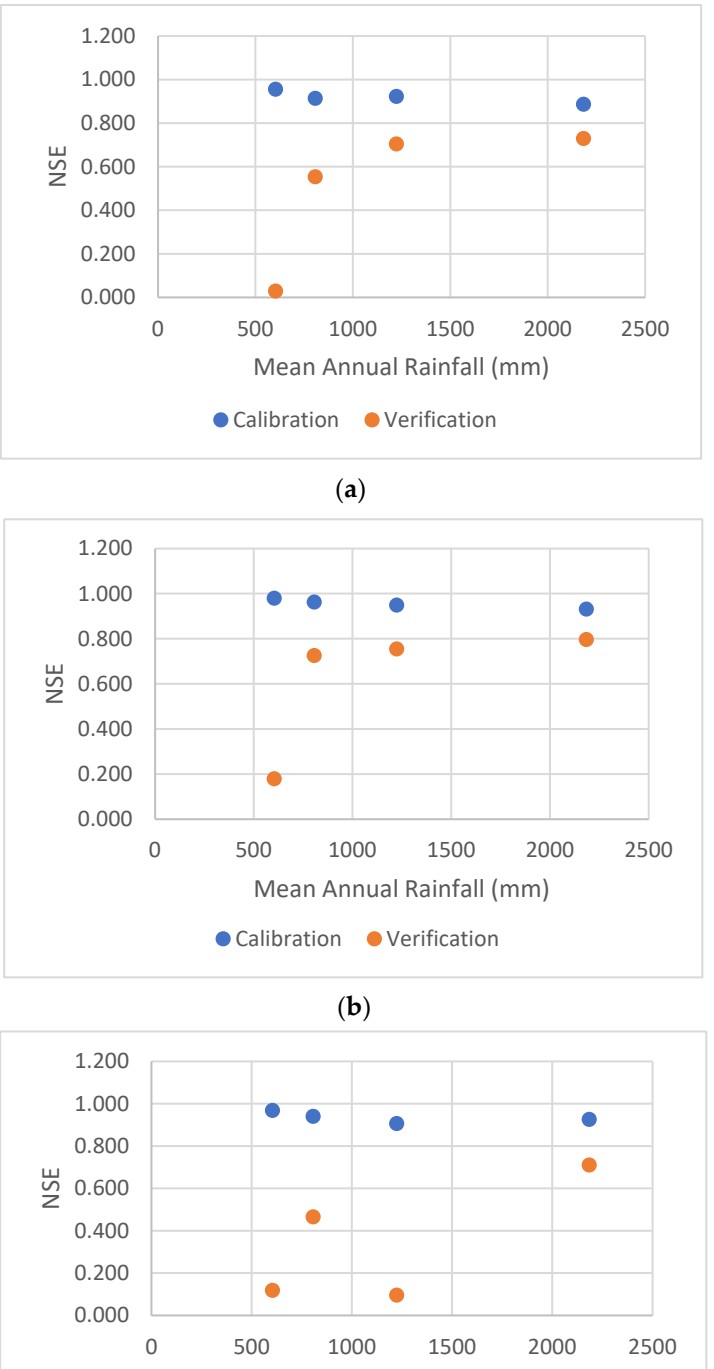

**Figure 9.** NSE of verification fits for the selected catchments—RORB (**a**), RRR (**b**), and HEC-HMS (**c**) models.

## 5. Conclusions

- All three models examined (RORB, RRR, and HEC-HMS) demonstrated satisfactory calibration performance on the selected catchments. However, the RRR model outperformed the others in terms of the overall level of fit, as indicated by the Nash–Sutcliffe Efficiency (NSE).

- The level of fit for the independent verification events, using weighted mean parameter values, was not as good as for the calibration events. Furthermore, there were variations in performance among the three models, with the RRR model exhibiting the highest NSE.
- The inclusion of baseflow estimation within the RRR model contributed to its superior performance in both calibration and verification, distinguishing it from the RORB and HEC-HMS models.
- The RRR model is capable of modelling extreme events involving more than two runoff processes, providing additional flexibility in flood estimation.
- The significance of climate change and the need for accurate catchment simulation by rainfall-based models supports a shift towards utilizing models such as the RRR model.
- These findings support the preference for multi-process models such as the RRR model over simpler models such as RORB and HEC-HMS which lack the capability to simulate multiple processes using runoff routing. Adopting multi-process models not only improves flood estimation but also allows for the simulation of extreme events involving additional processes.

It is recommended that catchment simulation using runoff routing models transition from single-process models such as RORB and HEC-HMS to multi-process models. Further research and integration of these models into standard engineering practices are necessary for their continued development and application.

**Author Contributions:** Conceptualization and methodology, D.K.; analysis, D.K. and G.H.A.; writing—original draft preparation, D.K.; writing—review and editing G.H.A. All authors have read and agreed to the published version of the manuscript.

**Funding:** This research received no external funding.

**Data Availability Statement:** There are no available data.

**Acknowledgments:** The authors would like to express their gratitude for the generous provision of data by the Department for Water and Environment, South Australia; the Department of Regional Development, Manufacturing and Water, Queensland; the Department of Water and Environmental Regulation, Western Australia; and Ecowise Environmental Pty. Ltd. Their contribution has been essential to the success of this study and is greatly appreciated.

**Conflicts of Interest:** The authors declare no conflict of interest.

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
