# Peer review of "Benchmarking Three Event-Based Rainfall-Runoff Routing Models on Australian Catchments"

_hydrology, doi:10.3390/hydrology10060131_

Round 1

Reviewer 1 Report

The article entitled “Benchmarking three event-based rainfall-runoff routing models on Australian catchments” addresses an important topic with potential applicability to other locations, mainly in watersheds with small and/or medium drainage areas. However, the authors should provide several corrections in the article, for future analysis.

- The INTRODUCTION is extremely “poor” regarding the study of the art of the “flood” theme. Displays only one citation. Authors must present a contextualization of the importance of this theme for Australia.

MATERIAL AND METHODS

- The section “MATERIAL AND METHODS” presents an inversion of reading flow, not being adequate. The following sequence is recommended: i) areas of study (characterization); ii) models; iii) statistical analysis. This will allow for a better understanding of the work.

- Use other statistical indicators (RMSE, MBE, MAE, d Willmott, among others). Only NSE is not enough to detail the statistical performance of estimation models.

- Why were only 6 events chosen for calibration and 6 for “verification”? Present the history of "floods" in the four hydrographic basins evaluated, for analysis of the representativeness of these 12 events. How many events were evaluated per basin?

- Figure 2 should present the hydrographic networks of each basin and their evaluation points, since some models are applied in sub-catchments. This Figure 2 must present N direction, grid of coordinates, among other technical Cartography information.

- Figure 3 should be better structured, mainly due to the adequacy of the double Y axis. Showing means and deviations for each variable and month.

RESULTS AND DISCUSSION

- The authors confuse “calibration” and “verification”, since they present statistical performance analyzes (NSE) for the two databases (Tables 3-4 and 5-6, respectively).

- Figures 5, 6 and 7 show NSE values on a scale of up to 1.2 (these values are not statistically possible) – Correct the figures. Based on these results, were the events well selected/representative by watershed? It seems that the tweaks in some cases were pretty bad.

- References are not adequate in terms of submission/template rules.

Author Response

Thanks for the comments.  The revised paper is a much better paper

Reviewer 2 Report

The authors have examined the performance of three models, for the prediction of flood flow in catchments. The authors have selected the techniques and assessed parameters effectively for evaluating the results relating to the objective of the study. I have observed a few flaws in the manuscript as highlighted below. 

Comments

1.    Introduction

Lines 35-38 – Include reference/s to support the idea given in the paragraph.

2. Materials and Methods

Line 101 – …. (at A, B, C, D and E in Figure 1) …. – add ‘Figure’ before 1. What about F and G?

Lines 118-119 – add a reference to Support the sentence.

Lines 187-188 – indicate whether the monthly rainfall, runoff, and potential areal evaporations are mean values.

Lines 190-192 – in line 192, check whether it should be ‘of’ instead ‘or’

Line 197 – Figure 3 - 

Figure 3 (a) – correct primary vertical axis – MMP and MMPE (mm) instead MMR and MMPE (mm).

Figure 3 (c) – Correct the primary vertical axis and show the secondary vertical axis title

Figure 3 (c) – horizontal axis – use the same format as in other figures, eg. Figure 3(a)

Line 205 – This was done using a 9 pass…… -  '9 pass' is it necessary? check and correct the sentence.

 Line 223-225 - …. Normalized by dividing ….?. -- check the sentence. It seems a parameter is missing.

Line 232 – peak flow ratio. Did you mean the weighting factor?

3.       Results

Figure 4 – in Legend – change HEC to HEC- HMS – be consistent with labels throughout the paper.

4.       Discussion

Line 288 - … independent events. (punctuation mark missing at the end of the sentence)

Figure 7 – check the title, NSE = HEC, if this correct?

Line 317 – The RRR model differs from the RRR …. – Please correct the sentence.

5.       Conclusion

Line 334 - … and, There is also … - check and correct the sentence.

Author Response

Thank you for your comments.  The revised paper has significant updates, and is a much better paper

Reviewer 3 Report

The manuscript is very well drafted and the topic is of very relevant subject with respect to the research of the day. However, there is a need to look into the following points

1. If the equations of the new model is closely match with the HEC- HMS, if that is case how the new model is different than that of the HEC-HMS. please explain in detail in the manuscript

2. Figure 2, The basic parameters are quite different for each of the catchment considered for the analysis. if there is a large variation in these parameters, how is possible to comparison between them. explain in detail

3. line 229 to line 231, it is mentioned that, the baseflow is extracted before the calibration and added back to the modelled hydrograph for comparison, if so which method of baseflow extraction is used. please provide the methodology

4.the results of the  RRR model is shown to be very good for event based model, is it possible to use this model for calibrating on daily basis.

keeping the above points, it may be recommended for minor corrections and may be accepted after the incorporation of the above points

good

Author Response

Thanks for your comments.  The revised paper has significant updates, and is a much better paper

Reviewer 4 Report

The techniques used and the quality of the data are appropriated to the question being studied. The conclusions are reliable. 

Author Response

Thanks for your review.  The paper has now been significantly updated as a result of comments from other reviewers.

Round 2

Reviewer 1 Report

The article entitled "Benchmarking three event-based rainfall-runoff routing models on Australian catchments" showed significant improvements in corrections, mainly regarding the characterization of models and watersheds evaluated. In a general analysis, it allows the issuance of a favorable opinion for the publication, subject to minor corrections.

- The APFE index (percentage absolute peak flow errors) must be presented in the sub-item "Statistical analysis" (lines 345-356);

- What is the meaning of the WF index (line 384)? At what point in the results and discussion do the authors address this proposed/evaluated index?

- The titles of tables and figures should be improved, in order to be self-explanatory. Eliminate identification of titles inside the graph/figure (example Fig. 8 - "Verification - NSE Rating"), as they must appear in the title of the Figure itself (if you have more than one graph per figure, these must be identified by letters ( A) or (B), and their identifications must appear in the title of the Figure.

- Figures 9, 10 and 11 can be grouped and identified as suggested above.

- The discussion item presents the citation of only one article/reference [8]. Are there no other articles with these models? Needs more in-depth discussion.

- The conclusions of the article must be rewritten in a direct and more objective way, responding to the "objectives" of the article. As it stands, it looks like discussion or "considerations".

Round 3

Reviewer 1 Report

The third revision of the article "Benchmarking three event-based rainfall-runoff routing models on Australian catchments" presents significant improvements and needs "small" corrections, which do not compromise the scientific content of the article.

- At several points in the text, there are paragraphs with only one line, characterizing weak scientific writing (example: line 327; line 429, which serves only to call Table 3 in the text).

- Table 4 is unnecessary, as this NSE classification is already consolidated and is duly referenced (line 435).

- Figure 9c, the letter "c" has the symbol "commercial"

- Line 539 is unnecessary
